# Deep Learning in Thoracic Oncology: Meta-Analytical Insights into Lung Nodule Early-Detection Technologies

**DOI:** 10.3390/cancers17040621

**Published:** 2025-02-12

**Authors:** Ting-Wei Wang, Chih-Keng Wang, Jia-Sheng Hong, Heng-Sheng Chao, Yuh-Min Chen, Yu-Te Wu

**Affiliations:** 1Institute of Biophotonics, National Yang-Ming Chiao Tung University, Taipei 30010, Taiwan; 2School of Medicine, College of Medicine, National Yang-Ming Chiao Tung University, Taipei 30010, Taiwan; 3Department of Computer Science, Whiting School of Engineering, Johns Hopkins University, Baltimore, MD 21218, USA; 4Department of Otolaryngology-Head and Neck Surgery, Taichung Veterans General Hospital, Taichung 40705, Taiwan; 5Department of Chest Medicine, Taipei Veteran General Hospital, Taipei 112, Taiwan; 6Brain Research Center, National Yang-Ming Chiao Tung University, Taipei 30010, Taiwan; 7College Medical Device Innovation and Translation Center, National Yang-Ming Chiao Tung University, Taipei 30010, Taiwan

**Keywords:** deep learning, lung nodule detection, convolutional neural networks, computed tomography scans, meta-analysis

## Abstract

Detecting lung nodules, which may indicate thoracic cancer, is a critical challenge in medical imaging. Recent advances in artificial intelligence, particularly deep learning, show promise for improving diagnostic accuracy. This study evaluates the effectiveness of deep learning models, such as convolutional neural networks, in identifying lung nodules on computed tomography images. By analyzing results from 48 studies, we aim to highlight the strengths and limitations of these models. Our findings emphasize the potential of artificial intelligence to transform lung cancer diagnosis while addressing the need for more inclusive and diverse studies to ensure these technologies are reliable and applicable across various patient populations. This work could help guide researchers, clinicians, and developers in advancing medical imaging solutions for cancer detection.

## 1. Introduction

Lung carcinoma, a leading cause of cancer-related deaths, results from the uncontrolled proliferation of abnormal cells in pulmonary tissues. Lung cancer is the most frequently diagnosed malignancy worldwide, with approximately 2.38 million new cases and 1.27 million deaths in 2023 [1]. The severe clinical symptoms of lung cancer considerably affect patients’ quality of life and place a substantial burden on health-care systems globally [2,3].

Despite advances in treatment strategies, the trend of diagnosing lung cancer at advanced stages remains a major challenge, adversely affecting survival [4]. For non-small-cell lung cancer, the 5-year survival rates are 65%, 37%, and 9% for localized, regional, and distant stages, respectively. By contrast, small-cell lung cancer has 5-year survival rates of 30%, 19%, and 3% for these stages, respectively [5]. Imaging modalities, such as computed tomography (CT), including its low-dose acquisition technique (LDCT), play pivotal roles in the early detection of lung cancer. The National Lung Screening Trial revealed the superiority of low-dose CT (LDCT) over conventional chest radiography, demonstrating a 20% reduction in mortality when LDCT was used for screening [6]. Furthermore, the NELSON trial revealed the efficacy of LDCT, and the results indicated a 26% reduction in mortality among men and an even more pronounced decrease of up to 61% in women compared with those who did not undergo screening [6]. These findings emphasize the importance of early detection for improving patient outcomes.

Despite the proven benefits of these imaging techniques, their integration into lung cancer diagnosis encounters some challenges. A major limitation is the delayed diagnosis of lung cancer, which is often detected only when advanced symptoms become apparent [7]. This delay in detection considerably reduces the effectiveness of early intervention strategies. In addition, these diagnostic methods are prone to high false-positive rates and the risk of overdiagnosis. These problems not only lead to unnecessary medical procedures and increased health-care costs but also cause considerable psychological distress among patients [8]. Thus, the integration of these diagnostic tools into clinical practice requires a careful evaluation of their limitations and their potential impact on patient health and health-care systems.

Substantial advancements have been made in lung cancer imaging in terms of artificial intelligence (AI), particularly in computer-aided detection (CADe) and computer-aided diagnosis (CADx). These developments are primarily reflected in the augmentation of detection capabilities [9], refined characterization of pulmonary nodules [10,11], and enhancement of risk assessment accuracy [12]. Among these advancements, a key development has been the integration of convolutional neural networks (CNNs) into the analysis of CT images. Empirical research has validated the effectiveness of CNNs in increasing the precision of detection and reducing the incidence of false positives [13]. This progress in AI and deep learning technologies represents a pivotal shift in the diagnostic landscape of lung cancer, increasing the accuracy and efficiency of early-detection methods.

Despite the promising role of CNNs, the inherent “black box” nature of these models limits their interpretability and explainability in direct end-to-end applications [14]. To address this problem, an alternative approach is to deconstruct the processes of CADe and CADx and subject them to critical analysis. Previous meta-analyses have primarily focused on the diagnostic accuracy of deep learning models [15,16,17]. This review examines the detection sensitivity of deep learning models for early lung cancer detection on CT images through CADe. We systematically synthesized and evaluated evidence on the role of AI in enhancing detection sensitivity and reducing the workload of radiologists. This review provides a comprehensive overview of the current state of AI in lung nodule detection and its potential implications for clinical practice, policy-making, and future research. Such an exploration can revolutionize the lung cancer screening process, optimizing early detection and thereby affecting patient outcomes.

## 2. Materials and Methods

### 2.1. General Guidelines

This systematic review and meta-analysis adhered to the Preferred Reporting Items for Systematic Reviews and Meta-Analyses (PRISMA) 2020 guidelines [18]. We followed the PRISMA checklist to ensure the highest methodological rigor in the design and reporting of this study. Detailed information regarding our compliance with these guidelines is provided in Appendix A. In addition, this study was registered with PROSPERO under CRD42023479887. Because this systematic review and meta-analysis did not involve direct human participation, ethical approval and informed consent requirements were deemed not applicable.

### 2.2. Database Search and Identification of Eligible Studies

Two reviewers (T-WW and J-SH) conducted a comprehensive literature review focusing on studies applying deep learning for chest CT images of lung cancer. The literature search was performed in PubMed, Embase, and Web of Science (Appendix A). The search period was from the inception of each database until 7 November 2023. Our selection process involved an initial screening of titles and abstracts and was supplemented by manual searches to ensure a comprehensive coverage of the relevant literature. Any discrepancies between reviewers during the selection phase were resolved through consultation with a third expert.

We included studies that recruited adults with lung cancer and used chest LDCT and CT images in conjunction with deep learning techniques. To maintain the focus and quality of our review, we excluded various study types and articles, including review articles, posters, conference presentations, and supplements. In addition, we excluded studies focusing on classification or false-positive reduction, articles not written in English, case reports, studies with dataset problems (e.g., incomplete dataset splitting), retracted articles, studies that did not use deep learning methodologies, studies that employed non-CT imaging methods, and those with irrelevant outcomes.

### 2.3. Data Extraction and Management

T-WW and J-SH conducted data extraction, focusing on various metrics relevant to lung nodule detection. Our investigative scope was expansive, covering various study characteristics, including study design, participants’ race, study duration, number of patients, data sources utilized, types and sizes of nodules, size thresholds, series and lesion numbers, lesion sizes, and methods employed for validation.

We evaluated the dimensions and types of algorithms employed in these studies. In addition, we examined various advanced preprocessing techniques, including generative methods, lung segmentation approaches, intensity standardization, resolution adjustments, image cropping, and data augmentation strategies. Furthermore, we analyzed false-positive reduction, multiscale approaches employed, model input dimensions, and specific deep learning models.

We analyzed the performance of deep learning algorithms, focusing on lesion-wise sensitivity. This performance metric was calculated to determine average lesion-wise sensitivity across an extensive range of false-positive thresholds per scan, including 1/8, 1/4, 1/2, 1, 2, 4, and 8. This metric also reflects precision, enabling a comprehensive assessment of the performance of deep learning models in the detection of lung nodules.

### 2.4. Methodological Quality Appraisal

The methodological quality of each study included in our review was thoroughly evaluated using the Checklist for Artificial Intelligence in Medical Imaging and the Quality Assessment of Diagnostic Accuracy Studies-2 tools [19,20]. T-WW and J-SH independently conducted this rigorous appraisal for the unbiased and comprehensive assessment of the studies. Given the complexities inherent in evaluating AI applications in medical imaging, any discrepancies were resolved through consultation with senior researchers. This step was crucial in harmonizing any conflicting interpretations and maintaining the integrity of the evaluation process.

### 2.5. Statistical Analysis

Two meta-analyses were performed to evaluate lesion-wise sensitivity, which was reported as a competition performance metric across the selected studies. The first analysis focused on independent datasets. When different studies utilized the same dataset, the highest-performing algorithm was chosen for inclusion. The second analysis included all reported results from all validation datasets. When studies reported multiple algorithms, the highest-performing algorithm was chosen for inclusion. For studies that presented median values and interquartile ranges, we converted them into mean values and standard deviations by using established formulas [21,22]. Given the heterogeneous study populations, we employed a random-effects model with restricted maximum likelihood (REML) [23], which offers more reliable estimates of between-study variance by adjusting for the degrees of freedom used in parameter estimation. The results of this comprehensive analysis are depicted in forest plots.

Subgroup analyses were conducted considering various factors, such as race (Asian vs. Western vs. mixed), nature of the database (open-source vs. private), validation methods (internal vs. external validation), dataset splitting methods (train/test split vs. cross-validation), and specific features of model architectures, including dimensionality (2D vs. 3D), image preprocessing procedures, false-positive reduction techniques, multiscale involvement, and types of algorithms [24]. Additionally, a meta-regression [25] was performed to explore correlations between competition performance metrics and variables such as the size of the training dataset and the number of lesions. The Q test was used to quantify discrepancies or heterogeneity across studies, with the statistical significance threshold set at *p* < 0.05. Heterogeneity was classified into the following categories based on I2 indices: trivial (0–25%), minimal (26–50%), moderate (51–75%), and pronounced (76–100%) [26]. To address potential biases in publication, Egger’s test was applied to assess funnel plot asymmetries [27].

## 3. Results

### 3.1. Study Identification and Selection

Our systematic literature review, depicted in the PRISMA flowchart (Figure 1), initially yielded 3485 studies. After removing 1255 duplicates, we screened 2330 articles by using EndNote. Of these, we excluded 2053 articles due to insufficient relevance. Further evaluation of 277 full-text articles led to additional exclusions for various reasons, such as the non-use of deep learning or the use of non-CT methods (Appendix A). Finally, we included 48 studies [28,29,30,31,32,33,34,35,36,37,38,39,40,41,42,43,44,45,46,47,48,49,50,51,52,53,54,55,56,57,58,59,60,61,62,63,64,65,66,67,68,69,70,71,72,73,74] in our comprehensive meta-analysis.

### 3.2. Basic Characteristics of Included Studies

Table 1 presents a comprehensive summary of patient and study characteristics from these 48 studies on lung cancer that were conducted from 2018 to 2023. These studies involved patient cohorts of varying sizes, with Zhao et al. (2023) [28] having the largest cohort of 4556 patients and Guo et al. (2020) [64] having the smallest cohort of 306 patients. Regarding lung cancer characterization, the studies primarily focused on nodules, often employing size thresholds of >3 mm. However, some studies [31,38,41,49,57,71] used different criteria. The lesion counts and sizes reported in the training and testing sets varied substantially, reflecting the heterogeneity of lung cancer. The validation methods employed in these studies mainly included training and testing frameworks, with several studies opting for cross-validation techniques. This observation highlights the importance of robust model validation in lung cancer research. Moreover, the consistent use of manual referencing across these studies emphasized the vital role of human expertise in ensuring data accuracy and contextual relevance, a key aspect of medical research.

### 3.3. Characteristics and Performance of Deep Learning Algorithms

Table 2 provides a comprehensive overview of the diverse methods used in lung cancer imaging research, highlighting the integration of various preprocessing techniques and deep learning algorithms. The data indicated a significant trend toward the use of generative algorithms, which were employed by several studies [19,33,57] for data augmentation. Lung segmentation, a crucial preprocessing step, was widely applied in many studies [28,29,30,31,33,34,35,36,37,38,39,40,41,42,43,47,49,50,51,52,53,54,55,56,57,59,64,67,69,71,72,73] for isolating regions of interest for accurate analysis. Intensity clipping [28,29,30,31,32,33,34,35,36,37,39,40,41,42,43,44,46,47,49,50,52,53,54,55,58,59,61,62,65,67,68,70] and normalization [28,29,30,31,32,33,34,35,36,39,40,41,42,43,44,46,47,48,49,50,51,52,53,54,55,58,59,60,61,62,65,67,68,70] were other critical preprocessing steps that were extensively implemented to ensure the standardization of imaging data across diverse datasets. Resolution adjustment [28,29,30,31,32,33,35,37,38,39,42,43,44,45,47,48,49,50,51,52,53,54,57,58,59,62,63,65,67,72,74] was also prominently used to ensure consistent image quality and details across different imaging modalities. Image augmentation [28,29,30,32,33,35,37,39,40,41,43,44,47,49,50,51,52,53,54,55,56,57,58,59,61,62,63,64,65,66,67,68,69,71,72,73] and cropping [28,29,30,31,32,33,34,35,36,37,38,39,40,41,42,43,44,45,46,47,48,49,51,52,53,54,55,56,57,58,59,60,61,62,63,64,65,66,67,68,69,71,72,73,74] were widely used to prevent overfitting and improve algorithmic robustness. Furthermore, false-positive reduction techniques [28,29,31,37,39,40,41,42,43,45,46,47,49,53,54,55,56,57,58,59,60,61,62,63,65,66,67,68,70,71,74] and multiscale analysis [29,30,34,39,40,41,42,46,47,49,51,52,53,54,56,58,60,62,63,65,66,67,70,73] were performed to enhance detection accuracy and reliability.

Different imaging dimensions, including 2D [28,37,55,58,59,67,70,71,74], 3D [29,30,31,33,34,35,36,38,39,40,41,42,43,44,45,46,47,48,49,50,51,52,53,54,56,57,60,61,62,63,64,65,67,68,69,73], and combined 2D/3D approaches [32,66], were used for lung nodule detection and characterization. Furthermore, these studies used diverse algorithms, ranging from basic U-net to advanced variants, such as U-net++, nnU-net, YOLOv5, Mask R-CNN, Faster R-CNN, and custom CNNs, reflecting the ongoing innovation and customized solutions for lung cancer detection. In these studies, the clinical performance measure values ranged from 0.39 to 0.96, indicating the different levels of efficacy of these approaches, which can be affected by dataset complexities and algorithmic capabilities. Additionally, Appendix A provides details on CT protocols and hardware characteristics, offering further insights into the imaging parameters used across studies.

### 3.4. Quality Assessment

Appendix A offers a comprehensive summary of the quality assessments for the studies included, utilizing the QUADAS-2 tool. Additionally, a detailed analysis focusing on bias risks and applicability concerns is presented in Appendix A.

Appendix A extensively evaluates 48 studies based on the CLAIM criteria. These studies yielded an average CLAIM score of 25.77, equating to approximately 61.36% of the maximum possible score of 42, with a standard deviation of 2.8 and scores ranging from 20.00 to 38.00. A breakdown of the average scores across CLAIM subsections for these studies is as follows: title/abstract section scored 1.7 out of 2 (85%), introduction 1.96 out of 2 (98%), methods 18.46 out of 28 (65.59%), results 1.38 out of 5 (27.5%), discussion 1.27 out of 2 (63.5%), and other information 1 out of 3 (33.3%). This breakdown highlights the varying levels of quality and thoroughness in different sections of the studies.

### 3.5. Efficacy of Deep Learning Model Detection of Lung Cancer on CT Images

An analysis of independent validation datasets for 16 deep learning algorithms revealed considerable variance in performance metrics, ranging from 39% to 96% (Figure 2). The mean performance metric was 79%, with a 95% confidence interval (CI) of 72% to 86%. These results revealed significant heterogeneity, as indicated by a Q-test value of 2214.72 (*p* < 0.01) and a Higgins I2 index of 99.18%. Sensitivity analysis (Appendix A) revealed markedly lower performance in the study by Xu et al. [60], leading to its exclusion from subsequent subgroup analysis and meta-regression. Egger’s test yielded a *p* value of 0.75, indicating no significant publication bias (Appendix A). Subgroup analyses revealed notable differences in performance for factors such as race, validation methods, resolution adjustment, and image cropping. However, variables such as dataset privacy, splitting methodology, slice thickness, generative approaches, lung segmentation, intensity clipping, intensity normalization, data augmentation, false-positive reduction strategies, multiscale analysis, model dimensions, and algorithm type did not significantly affect the performance of these models (Figure 3). The analysis including Xu et al. is presented in Appendix A. In addition, meta-regression analysis did not reveal a significant correlation between the number of training images or lesion counts and the overall effectiveness of the models.

An extensive evaluation of validation datasets for 64 deep learning algorithms revealed substantial variability in performance metrics, ranging from 39% to 96% (Figure 4). The calculated average performance metric was 85%, with a 95% CI ranging from 83% to 88%. This substantial heterogeneity was supported by a Q-test value of 7133.19 (*p* < 0.01) and a Higgins I2 index of 99.23%. Sensitivity analysis (Appendix A) did not reveal a significant difference based on the exclusion of any article. Egger’s test yielded a *p* value of <0.01, indicating the presence of publication bias (Appendix A). However, additional subgroup analysis incorporating conference papers did not reveal significant differences (Appendix A). Subgroup analysis demonstrated significant differences in model performance for race, dataset privacy, and dataset splitting methods. However, no significant differences were observed in model performance for the validation methods, slice thicknesses, generative approaches, lung segmentation, intensity clipping, intensity normalization, resolution adjustment, data augmentation, image cropping, false-positive reduction strategies, multiscale analysis, model dimensions, or algorithm type (Figure 5). In addition, meta-regression analysis indicated no significant correlations between the number of training images or lesion counts and the overall efficacy of the models.

## 4. Discussion

To the best of our knowledge, this is the first systematic review and meta-analysis to evaluate the sensitivity of deep learning models for lung nodule detection using competition performance metrics as a benchmark. Unlike previous studies that primarily focused on diagnostic sensitivity and specificity [75] or segmentation performance using metrics such as the Dice similarity coefficient [76], our study uniquely synthesizes results from 16 deep learning models validated on independent datasets. Through this analysis, we derived a pooled competition performance metric of 79%, with a 95% confidence interval ranging from 72% to 86%, providing a novel perspective on the evaluation of model performance in this domain. Race, validation methods, resolution adjustment, and image cropping significantly affected the performance of these deep learning models. Furthermore, when examining the results for all 64 deep learning algorithms across various validation datasets, the pooled competition performance metric was 85%, with 95% CI values ranging from 83% to 88%. Race, dataset privacy, and dataset splitting methods significantly affected the performance of these deep learning algorithms.

This study revealed the significant effect of race on the performance of deep learning models. Notably, we observed that a higher proportion of individuals of Asian descent exhibited non-solid nodules with a ground-glass opacity appearance [77,78]. This characteristic might have contributed to the decreased performance, either due to the non-solid morphology of nodules or a potential imbalance in distribution within the dataset, leading to reduced sensitivity. Furthermore, we identified that image distortions differentially affected the performance of deep learning models across various subgroups. This suggests the increased sensitivity of these models to variations in input data [79]. Such increased sensitivity might contribute to the substantial effect of race on model performance. A recent study [80] highlighted the importance of fairness and equity in deep learning models, advocating for equitable model performance across different subpopulations in datasets, particularly for variables such as race and gender. Thus, deep learning algorithms that are not only effective but also impartial and inclusive should be developed, considering the diversity of the populations they are intended to serve.

Our analysis of validation methods revealed that external validation exhibited superior performance compared to internal validation, a finding that contrasts with the commonly observed decrease in performance under external validation [81]. One possible explanation is that the external datasets used in this study may have been either more representative of real-world conditions or less heterogeneous, allowing the models to generalize effectively. In addition, robust training procedures, such as appropriate data augmentation and thorough hyperparameter tuning, could have contributed to reducing overfitting and improving model resilience when evaluated on external data. Consequently, this outcome suggests that well-designed deep learning models can maintain high performance across diverse populations, highlighting their potential robustness. However, when the analysis was extended to larger sample sizes across all validation sets, no significant difference was identified between internal and external validation methods. Furthermore, dataset privacy and dataset splitting methods significantly affect model performance. Open-source datasets, such as LUNA16 [74], Tianchi [82], Russia [83], Pn9 [46], SPIE-AAPM Lung CT Challenge [84], Lung Time [85], LNDb [86], and NLST [87], have been extensively utilized for benchmarking various deep learning algorithms. Notably, the LUNA16 dataset consists of low-dose CT (LDCT) scans, whereas several other studies included in this review relied on full-dose CT scans from private datasets. This distinction is important, as LDCT is commonly used for lung cancer screening due to its reduced radiation exposure, but its impact on model performance remains underexplored. Although cross-validation can enhance the representativeness of validation results, it has a risk of overfitting. The findings collectively indicate the significant impact of dataset characteristics on the performance of deep learning models. They highlight the need for further research for validating these models in diverse clinical settings and with varied dataset partitions. Such research would ensure the applicability and reliability of deep learning models across different clinical environments and patient populations, thereby contributing to the development of more robust and universally applicable diagnostic tools in the health-care sector.

In terms of dataset preprocessing, we determined that resolution adjustment and image cropping were significantly correlated with the performance of deep learning models. However, less than five studies focused on this aspect, potentially leading to a lack of statistical power. Further analysis incorporating a broader range of validation datasets did not corroborate these initial findings. Despite the inconclusiveness regarding resolution adjustment and image cropping, the results revealed that preprocessing enhanced model performance. Regarding the algorithms used in deep learning models, segmentation-based U-Net architectures and their variants exhibited higher performance than detection-based CNNs and other customized CNN architectures, aligning with findings from a recent study [28]. A possible explanation is that U-Net’s encoder–decoder structure with skip connections may facilitate more precise localization of nodules by preserving spatial information and capturing subtle textural details. In contrast, detection-based CNNs often focus on bounding-box predictions and may therefore miss fine-grained distinctions crucial for small or irregularly shaped nodules. Nonetheless, we did not observe any statistically significant differences in performance across these algorithm types. Furthermore, other model-related factors, such as slice thickness, false-positive reduction strategies, multiscale analysis, model dimensions, and various preprocessing techniques, did not exhibit significant differences among subgroups.

The study results revealed that the training size and lesion count were not significantly correlated with model performance. This finding is consistent with those of previous studies [79,88,89]. These observations suggest that dataset characteristics exert a more significant effect on the performance of deep learning models than technical factors related to deep learning. This highlights the importance of focusing on dataset characteristics, such as quality, diversity, and representativeness, for enhancing the efficacy of deep learning models in clinical and research applications.

This study demonstrated the potential and inherent challenges of employing deep learning models, especially CNNs, for early lung cancer detection. These models have demonstrated their efficacy in enhancing the precision of pulmonary nodule detection on CT images, improving diagnostic accuracy, and alleviating the workload for radiologists. However, the clinical application of these technologies is more complex. A critical limitation of the included studies is their non-interventional nature, indicating that AI assessments did not affect the diagnostic workflow. Thus, whether the enhanced cancer detection capabilities of these models in research would effectively translate into actual improvements in cancer detection in real-world clinical settings remains to be determined.

Translating deep learning-based lung cancer detection into clinical practice requires a comprehensive, multistep approach. Prospective clinical trials are essential to validate model performance under real-world conditions, ensuring generalizability and robustness across diverse patient populations and workflows. Effective integration into radiology workflows must include training staff, ensuring user-friendly interfaces, and achieving seamless compatibility with existing systems. Adhering to regulatory approvals is crucial to ensure safety and reliability, while implementing robust data privacy and security protocols is necessary to protect patient information and comply with regulations. Furthermore, interpretability and explainability are vital to foster trust and acceptance, enabling clinicians to understand and rely on AI-driven decisions. Lastly, evaluating the cost-effectiveness of these technologies will inform resource allocation and reimbursement strategies. These steps together provide a clear roadmap for bringing deep learning solutions into routine clinical use.

In future work, differentiating between commercially available and non-commercially available AI software could provide valuable insights, as highlighted in the study by Fukumoto et al. [90], which validated the performance of commercially available deep-learning-based lung nodule detection using Synapse Fujifilm. The “black box” nature of many DL models raises concerns regarding their interpretability and reliability in clinical settings. Performance variability based on training dataset characteristics, including racial diversity, can lead to biases. This study revealed performance discrepancies across different datasets, emphasizing the need for extensive validation in varied clinical environments. Future larger-scale prospective or interventional studies should focus on developing practical, fair, inclusive, and interpretable models to integrate DL in thoracic oncology effectively, ensuring their broad applicability across diverse patient populations.

## 5. Conclusions

This study effectively demonstrated the capability of deep learning models, especially CNNs, in enhancing the detection of lung nodules on CT images. The pooled competition performance metrics of 79% (95% CI: 72% to 86%) on independent datasets and 85% (95%CI: 83% to 88%) on all validation sets indicate the potential of these technologies to substantially improve diagnostic accuracy and reduce the workload of radiologists. However, despite these promising results, several critical challenges exist that must be addressed. These include the non-interventional nature of the studies, the “black box” problem inherent in AI models, and variability in performance based on dataset characteristics. Highlighting the need for extensive, diverse, and interventional research, the study emphasizes developing effective, interpretable, and equitable deep learning models across different patient demographics. This approach is crucial for advancing lung cancer screening and diagnosis in clinical practice.

## Figures and Tables

**Figure 1 cancers-17-00621-f001:**
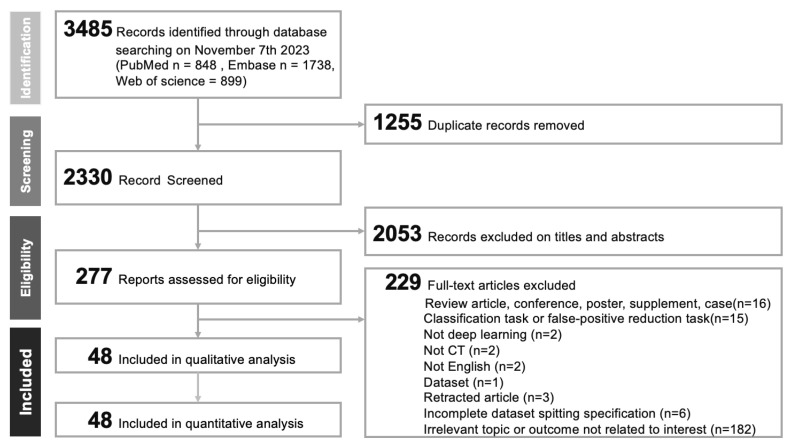
PRISMA flowchart for study selection.

**Figure 2 cancers-17-00621-f002:**
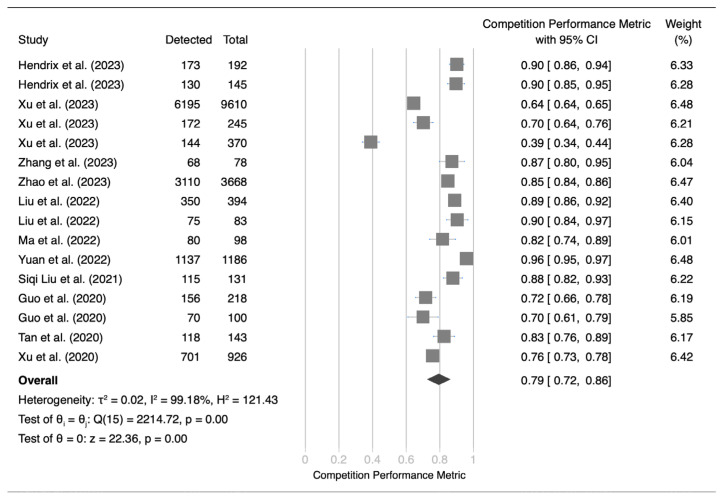
Forest plot of competition performance metrics of deep learning algorithms in independent validation datasets [28,29,31,37,43,44,47,49,52,60,61].

**Figure 3 cancers-17-00621-f003:**
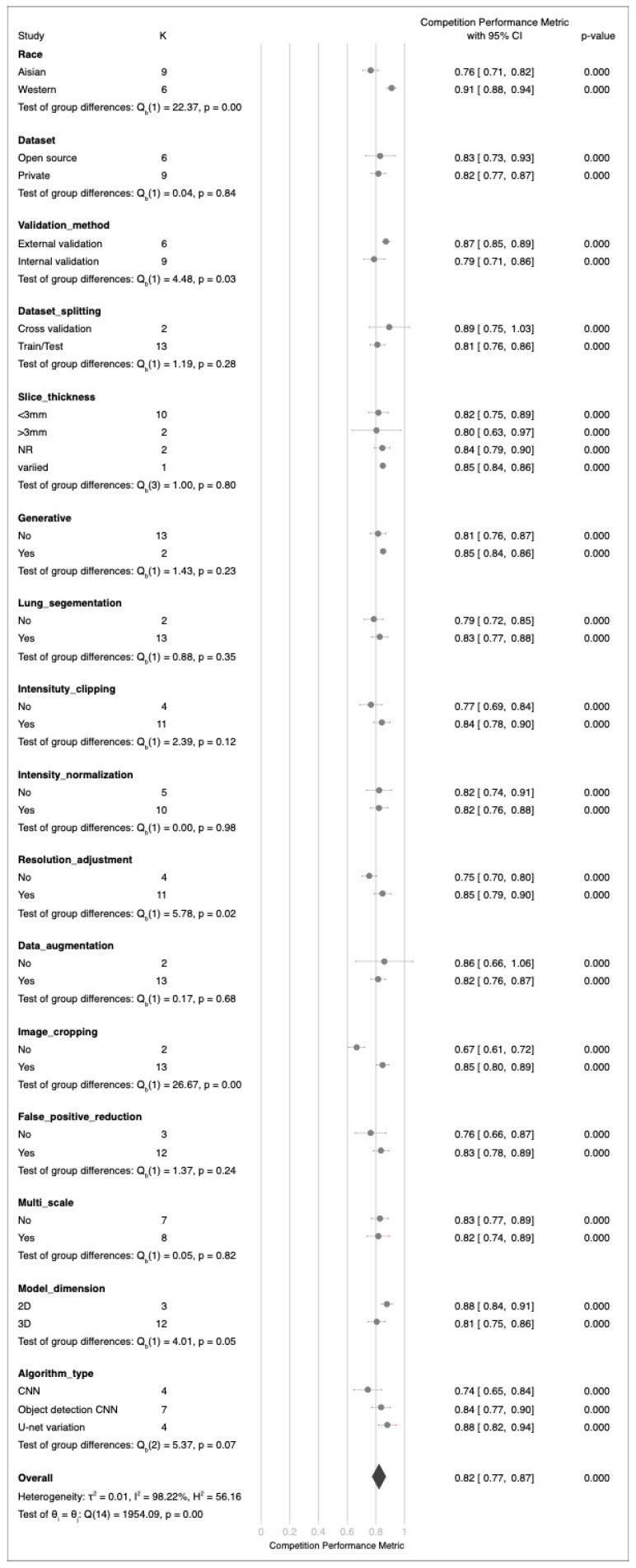
Forest plot of subgroup analysis competition performance metrics of deep learning algorithms in independent datasets.

**Figure 4 cancers-17-00621-f004:**
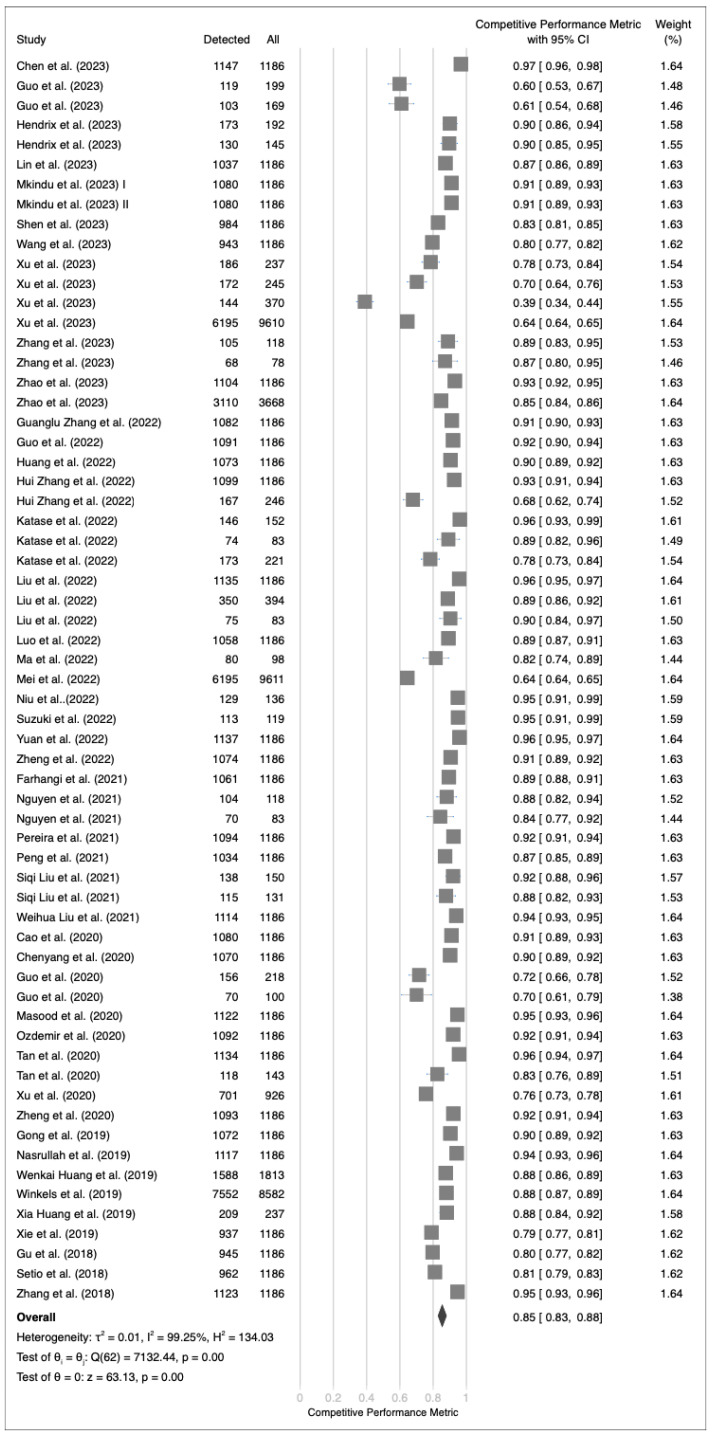
Forest plot of competition performance metrics of deep learning algorithms in all validation datasets [28,29,30,31,32,33,34,35,36,37,38,39,41,42,43,44,45,46,47,48,49,50,51,52,53,54,55,56,57,58,59,61,62,63,65,66,67,68,69,70,71,72,73,74].

**Figure 5 cancers-17-00621-f005:**
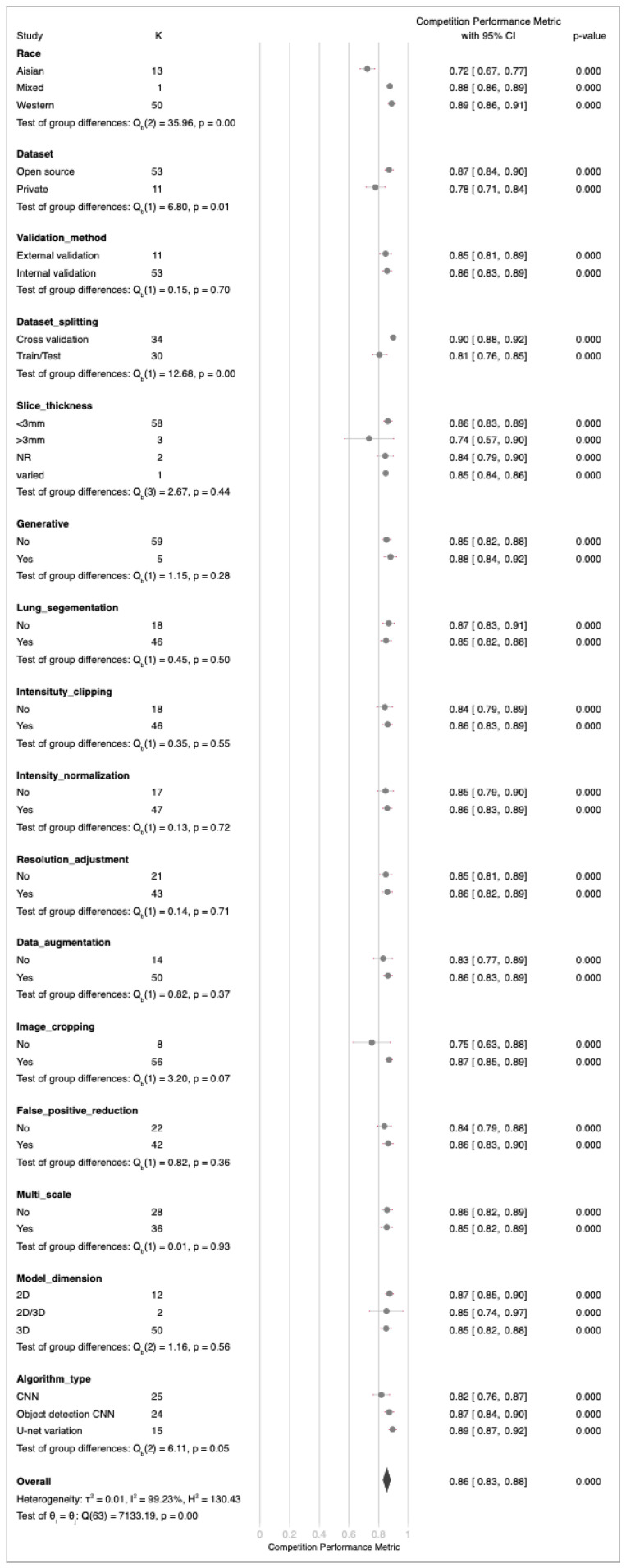
Forest plot of subgroup analysis competition performance metrics of deep learning algorithms in all validation datasets.

**Table 1 cancers-17-00621-t001:** Patient and study characteristics.

First Author	Publication Year	Study Design	Race	Patients	Data Source	Image	Duration	Type	Size Threshold	Series (Train/Test)	Lesion Num (Train/Test)	Size (Train/Test)	Validation
Zhao et al. [28]	2023	Retrospective	I: Western; E: Asian	4556	I: LUNA16, E: Shanghai Chest Hospital	LDCT/CT	2012~2018	I: nodule; E: malignancy	LUNA16 > 3 mm	4556 (888/3668)	4854 (1186/3668)	NR (8.31/15.17)-mean	Train/test
Zhang et al. [29]	2023	Retrospective	Western	888	LUNA 16	LDCT	NR	Nodule	LUNA16 > 3 mm	888 (710/178)	1186 (949/237)	8.31 (NR/NR)-mean	Train/test
Zhang et al. [30]	2023	Retrospective	I: Western E: Asian	928	I: LUNA16, E: Wuhan Hospital	LDCT/CT	NR	Nodule	LUNA16 > 3 mm	928 (800/128)	1264 (1068/196)	NR (8.31/NR)-mean	Train/test
Xu et al. [31]	2023	Retrospective	I1: Western; I2: Asian; I3: Western; I4: Asian	10,848	I1: LUNA16; I2: Tianchi; I3: Russia; I4: PN9	LDCT/CT	2015~2019	Nodule	LUNA16 > 3 mm; Tianchi > 5 mm; Russia > 3 mm	10,848 (8349/2499)	44,719 (34,252/10,467)	NR (8.31/NR/NR/NR)-mean	Train/test
Wang et al. [32]	2023	Retrospective	Western	888	LUNA 16	LDCT	NR	Nodule	LUNA16 > 3 mm	888 (799/89)	1186 (1067/119)	8.31 (NR/NR)-mean	Cross-validation
Shen et al. [33]	2023	Retrospective	Western	888	LUNA 16	LDCT	NR	Nodule	LUNA16 > 3 mm	888 (799/89)	1186 (1067/119)	8.31 (NR/NR)-mean	Cross-validation
Mkindu et al. I [34]	2023	Retrospective	Western	888	LUNA 16	LDCT	NR	Nodule	LUNA16 > 3 mm	888 (799/89)	1186 (1067/119)	8.31 (NR/NR)-mean	Cross-validation
Mkindu et al. II [35]	2023	Retrospective	Western	888	LUNA 16	LDCT	NR	Nodule	LUNA16 > 3 mm	888 (799/89)	1186 (1067/119)	8.31 (NR/NR)-mean	Cross-validation
Lin et al. [36]	2023	Retrospective	Western	888	LUNA 16	LDCT	NR	Nodule	LUNA16 > 3 mm	888 (799/89)	1186 (1067/119)	8.31 (NR/NR)-mean	Cross-validation
Hendrix et al. [37]	2023	Retrospective	Western	1690	LUNA 16, Radboud University Medical Center, Jeroen Bosch Hospital	LDCT/CT	2017~2020	Nodule	LUNA16 > 3 mm	1690 (1490/200)	5107 (4770/337)	NR	Train/test
Guo et al. [38]	2023	Retrospective	I1: Asian; I2: Asian	184	West China Hospital	LDCT	2008~2017	Nodule	1 mm~30 mm	578 (459/119)	1781 (1314/467)	NR (8.43/7.67)-mean	Train/test
Chen et al. [39]	2023	Retrospective	Western	888	LUNA 16	LDCT	NR	Nodule	LUNA16 > 3 mm	888 (799/89)	1186 (1067/119)	8.31 (NR/NR)-mean	Cross-validation
Zheng et al. [40]	2022	Retrospective	Western	888	LUNA 16	LDCT	NR	Nodule	LUNA16 > 3 mm	888 (799/89)	1186 (1067/119)	8.31 (NR/NR)-mean	Cross-validation
Hui Zhang et al. [41]	2022	Retrospective	Western	1088	I1: LUNA16; I2: Tianchi	LDCT	NR	Nodule	LUNA16 > 3 mm; Tianchi > 5 mm	1088 (799/289)	1432 (1067/365)	NR (8.31/NR)-mean	Train/test
Guanglu Zhang et al. [42]	2022	Retrospective	Western	888	LUNA 16	LDCT	NR	Nodule	LUNA16 > 3 mm	888 (799/89)	1186 (1067/119)	8.31 (NR/NR)-mean	Cross-validation
Yuan et al. [43]	2022	Retrospective	Western	888	LUNA 16	LDCT	NR	Nodule	LUNA16 > 3 mm	888 (799/89)	1186 (1067/119)	8.31 (NR/NR)-mean	Cross-validation
Suzuki et al. [44]	2022	Retrospective	Western	888	LUNA 16	LDCT	NR	Nodule	LUNA16 > 3 mm	888 (799/89)	1186 (1067/119)	8.31 (NR/NR)-mean	Cross-validation
Niu et al. [45]	2022	Retrospective	Western	436	LUNA 16	LDCT	NR	Nodule	LUNA16 > 6 mm	436 (349/87)	684 (548/136)	NR	Train/test
Mei et al. [46]	2022	Retrospective	Asian	8798	PN9	CT	2015~2019	Nodule	NR	8798 (6707/2091)	40,439 (30,828/9611)	NR	Train/test
Ma et al. [47]	2022	Retrospective	Asian	456	First Hospital of China Medical University	CT	NR	GGO	NR	456 (365/91)	486 (388/98)	NR	Cross-validation
Luo et al. [48]	2022	Retrospective	Western	888	LUNA 16	LDCT	NR	Nodule	LUNA16 > 3 mm	888 (799/89)	1186 (1067/119)	8.31 (NR/NR)-mean	Cross-validation
Liu et al. [49]	2022	Retrospective	I: Western; E1: Western; E2: Western	1118	LUNA 16; SPIE-AAPM; Lung TIME	LDCT/CT	NR	Nodule	LUNA16 > 3 mm, SPIE-AAPM > 10 mm Lung TIME > 2 mm	1118 (799/319)	1663 (1067/596)	NR (8.31/NR/NR)-mean	Train/test
Katase et al. [50]	2022	Retrospective	I: Mixed; E1: Western; E2: Western	2374	Kyorin University Hospital; LUNA 16; SPIE-AAPM; LNDb	LDCT/CT	2013~2018	Nodule	Major axis diameter measured ≥ 3 mm for solid or part-solid nodules and ≥5 mm for ground-glass nodules	2374 (1997/377)	10,924 (10,467/457)	NR (8.3/NR/NR)-mean	Train/test
Huang et al. [51]	2022	Retrospective	Western	888	LUNA 16	LDCT	NR	Nodule	LUNA16 > 3 mm	888 (710/178)	1186 (949/237)	8.31 (NR/NR)-mean	Cross-validation
Guo et al. [52]	2022	Retrospective	Western	888	LUNA 16	LDCT	NR	Nodule	LUNA16 > 3 mm	888 (799/89)	1186 (1067/119)	8.31 (NR/NR)-mean	Cross-validation
Pereira et al. [53]	2021	Retrospective	Western	888	LUNA 16	LDCT	NR	Nodule	LUNA16 > 3 mm	888 (799/89)	1186 (1067/119)	8.31 (NR/NR)-mean	Cross-validation
Peng et al. [54]	2021	Retrospective	Western	888	LUNA 16	LDCT	NR	Nodule	LUNA16 > 3 mm	888 (799/89)	1186 (1067/119)	8.31 (NR/NR)-mean	Cross-validation
Nguyen et al. [55]	2021	Retrospective	Western	961	LUNA 16	LDCT	NR	Nodule	LUNA16 > 3 mm	961 (799/162)	1269 (1068/201)	8.31 (NR/NR)-mean	Cross-validation
Weihua Liu et al. [56]	2021	Retrospective	Western	888	LUNA 16	LDCT	NR	Nodule	LUNA16 > 3 mm	888 (799/89)	1186 (1067/119)	8.31 (NR/NR)-mean	Cross-validation
Siqi Liu et al. [57]	2021	Retrospective	Western	6904	LUNA 16, NLST, in-house	LDCT/CT	NR	Nodule	> 6 mm	6904 (6488/446)	22,450 (22,169/281)	NR	Train/test
Farhangi et al. [58]	2021	Retrospective	Western	888	LUNA 16	LDCT	NR	Nodule	LUNA16 > 3 mm	888 (799/89)	1186 (1067/119)	8.31 (NR/NR)-mean	Cross-validation
Zheng et al. [59]	2020	Retrospective	Western	888	LUNA 16	LDCT	NR	Nodule	LUNA16 > 3 mm	888 (799/89)	1186 (1067/119)	8.31 (NR/NR)-mean	Cross-validation
Xu et al. [60]	2020	Retrospective	Asian	1109	The First Affiliated Hospital of Nanjing Medical University	CT	2014~2015	Nodule	NR	1109 (998/111)	12,057 (11,131/926)	NR	Train/test
Tan et al.. [61]	2020	Retrospective	I: Western; E: Asian	1003	I: LUNA 16; E: Zhejiang Cancer Hospital,	LDCT/CT	NR	Nodule	LUNA16 > 3 mm	1003 (799/204)	1329 (1067/262)	NR (8.31/NR)-mean	Train/test
Ozdemir et al. [62]	2020	Retrospective	Western	888	LUNA 16	LDCT	NR	Nodule	LUNA16 > 3 mm	888 (799/89)	1186 (1067/119)	8.31 (NR/NR)-mean	Cross-validation
Masood et al. [63]	2020	Retrospective	Western	888	LUNA 16	LDCT	NR	Nodule	LUNA16 > 3 mm	888 (799/89)	1186 (1067/119)	8.31 (NR/NR)-mean	Cross-validation
Guo et al. [64]	2020	Retrospective	Asian	306	West China Hospital	LDCT	NR	Nodule	NR	579 (463/116)	1590 (1272/318)	NR	Train/test
Chenyang et al. [65]	2020	Retrospective	Western	601	LUNA 16	LDCT	NR	Nodule	LUNA16 > 3 mm	888 (799/89)	1186 (1067/119)	8.31 (NR/NR)-mean	Cross-validation
Cao et al. [66]	2020	Retrospective	Western	601	LUNA 16	LDCT	NR	Nodule	LUNA16 > 3 mm	888 (799/89)	1186 (1067/119)	8.31 (NR/NR)-mean	Cross-validation
Xie et al. [67]	2019	Retrospective	Western	601	LUNA 16	LDCT	NR	Nodule	LUNA16 > 3 mm	888 (710/178)	1186 (949/237)	8.31 (NR/NR)-mean	Cross-validation
Winkels et al. [68]	2019	Retrospective	Western	NR	I: NLST, E: LIDC/IDRI	LDCT	NR	Nodule	NR	NR	47,462 (38,889/8582)	NR (NR/8.31)-mean	Train/test
Nasrullah et al. [69]	2019	Retrospective	Western	601	LUNA 16	LDCT	NR	Nodule	LUNA16 > 3 mm	888 (710/178)	1186 (949/237)	8.31 (NR/NR)-mean	Cross-validation
Xia Huang et al. [70]	2019	Retrospective	Western	601	LUNA 16	LDCT	NR	Nodule	LUNA16 > 3 mm	888 (710/178)	1186 (949/237)	8.31 (NR/NR)-mean	Train/test
Wenkai Huang et al. [71]	2019	Retrospective	Western + Asian	NR	LUNA16, Tianchi	LDCT/CT	NR	Nodule	LUNA16 > 3 mm; Tianchi > 5 mm	1888 (1699/189)	18,125 (16,312/1813)	NR	Cross-validation
Gong et al. [67]	2019	Retrospective	Western	601	LUNA 16	LDCT	NR	Nodule	LUNA16 > 3 mm	888 (799/89)	1186 (1067/119)	8.31 (NR/NR)-mean	Cross-validation
Zhang et al. [72]	2018	Retrospective	Western	601	LUNA 16	LDCT	NR	Nodule	LUNA16 > 3 mm	888 (799/89)	1186 (1067/119)	8.31 (NR/NR)-mean	Cross-validation
Gu et al. [73]	2018	Retrospective	Western	601	LUNA 16	LDCT	NR	Nodule	LUNA16 > 3 mm	888 (799/89)	1186 (1067/119)	8.31 (NR/NR)-mean	Cross-validation
Setio et al. [74]	2018	Retrospective	Western	601	LUNA 16	LDCT	NR	Nodule	LUNA16 > 3 mm	888 (799/89)	1186 (1067/119)	8.31 (NR/NR)-mean	Cross-validation

**Table 2 cancers-17-00621-t002:** Characteristics and performance of preprocessing techniques and deep learning algorithms.

First Author	Training Size	Lesion Count	Generative	Lung Segmentation	Intensity Clipping	Intensity Normalization	Resolution Adjustment	Image Augmentation	Image Cropping	False-Positive Reduction	Multiscale	Input Dimension	Algorithms	CPM
Zhao et al. [28]	888	1186	Yes	Yes	−1024, 400	−1, 1	1 × 1 × 1	Yes	64 × 64 × 64	Yes	No	2D	UNet++	E: 0.848
Zhang et al. [29]	710	949	No	Yes	−1200, 600	0, 255	1 × 1 × 1	Yes	128 × 128 × 128	Yes	Yes	3D	FPN	0.8934
Zhang et al. [30]	800	1068	No	Yes	−1200, 600	0, 255	Yes	Yes	128 × 128 × 128	No	Yes	3D	SK-ResNet	E: 0.8742
Xu et al. [31]	8349	34252	No	Yes	−1200, 600	0, 255	1 × 1 × 1	No	128 × 128 × 128	Yes	No	3D	NoduleNet+SGSE	I1: 0.7830; I2: 0.7036; I3:0.3901; I4:0.6446
Wang et al. [32]	799	1067	No	No	−1000, 400	0, 1	Yes	Yes	50 × 50 × 50	No	No	2D/3D	CNN	0.795
Shen et al. [33]	799	1067	Yes	Yes	−1200, 600	0, 255	Yes	Yes	96 × 96 × 96	No	No	3D	CADe	0.8299
Mkindu et al. I [34]	799	1067	No	Yes	−1200, 600	0, 1	No	No	64 × 64 × 64, 16 × 16 × 16, 12 × 12 × 12	No	Yes	3D	MSViT	0.911
Mkindu et al. II [35]	799	1067	No	Yes	−1200, 600	0, 1	1 × 1 × 1	Yes	64 × 64 × 64	No	No	3D	ResECA	0.911
Lin et al. [36]	799	1067	No	Yes	−1200, 600	0, 1	No	No	96 × 96 × 96	No	No	3D	IR-UNet++	0.8741
Hendrix et al. [37]	1490	4770	No	Yes	−1000, 400	No	Yes	Yes	512 × 512	Yes	No	2D	YOLOv5	0.902
Guo et al. [38]	459	1314	No	Yes	No	No	Yes	No	128 × 128 × 20	No	No	3D	CNN	I1: 0.6/I2:0.6116
Chen et al. [39]	799	1067	No	Yes	−1000, 400	0, 1	1 × 1 × 1	Yes	64 × 64 × 12	Yes	Yes	3D	F-Net, MSS-Net	0.967
Zheng et al. [40]	799	1067	No	Yes	−1200, 600	0, 255	No	Yes	210 × 210 × 210	Yes	Yes	3D	TSND	0.9059
Hui Zhang et al. [41]	799	1067	No	Yes	−1200, 600	0, 255	No	Yes	96 × 96 × 96	Yes	Yes	3D	Faster R-CNN	0.927
Guanglu Zhang et al. [42]	799	1067	No	Yes	−1200, 600	0, 255	1 × 1 × 1	No	128 × 128 × 128	Yes	Yes	3D	FPN+CSA	0.912
Yuan et al. [43]	799	1067	No	Yes	−1200, 600	0, 255	1 × 1 × 1	Yes	48 × 48 × 48	Yes	Yes	3D	3D Residual U-Net model	0.959
Suzuki et al. [44]	799	1067	No	No	−1000, 600	0, 255	1 × 1 × 1	Yes	64 × 96 × 96	No	No	3D	Modified 3D U-net	0.947
Niu et al. [45]	349	548	No	No	No	No	Yes	No	96 × 96 × 96	Yes	No	3D	URCTrans	0.95
Mei et al. [46]	6707	30828	No	No	−1200, 600	0, 255	No	No	128 × 128 × 128	Yes	Yes	3D	SANet	0.6446
Ma et al. [47]	365	388	No	Yes	−300, 1800	Yes	Yes	Yes	64 × 128 × 128	Yes	Yes	3D	Mask RCNN	0.8174
Luo et al. [48]	799	1067	No	No	No	0, 1	1 × 1 × 1	No	96 × 96 × 96	No	No	3D	SCPM-Net	0.892
Liu et al. [49]	799	1067	No	Yes	−1200, 600	0, 255	1 × 1 × 1	Yes	96 × 96 × 96	Yes	Yes	3D	FPN	0.957
Katase et al. [50]	1997	10467	No	Yes	Yes	0–1	1 × 1 × 1	Yes	No	No	No	3D	Faster R-CNN	0.96
Huang et al. [51]	710	949	No	Yes	No	0–1	1 × 1 × 1	Yes	80 × 80 × 80	No	Yes	3D	OSAF-YOLOv3	0.905
Guo et al. [52]	799	1067	No	Yes	−1200, 600	0, 255	No	No	96 × 96 × 96	No	Yes	3D	Multiscale aggregation network	0.92
Pereira et al. [53]	799	1067	No	Yes	−1000, 400	0, 255	Yes	Yes	32 × 32 × 32, 48 × 48 × 48	Yes	Yes	3D	Mask R-CNN	0.9224
Peng et al. [54]	799	1067	No	Yes	−1200, 600	0–1	1 × 1 × 1	No	128 × 128 × 128	Yes	Yes	3D	CEM, SAM, and Res2Net	0.872
Nguyen et al. [55]	799	1068	No	Yes	−1200, 600	0–1	No	Yes	512 × 512	Yes	No	2D	Faster R-CNN	0.882
Weihua Liu et al. [56]	799	1067	No	Yes	No	No	No	Yes	128 × 128 × 128	Yes	Yes	3D	PDCB	0.939
Siqi Liu et al. [57]	6488	22169	Yes	Yes	No	No	Yes	Yes	64 × 64 × 64	Yes	No	3D	Deep Lung	0.9226
Farhangi et al. [58]	799	1067	No	No	−1000, 400	0–1	0.625 × 0.625 × 2	Yes	48 × 48 × 16	Yes	Yes	2D	CNN	0.895
Zheng et al. [59]	799	1067	No	Yes	−1000, 400	0–1	Yes	Yes	512 × 512	Yes	No	2D	DL-CAD	0.922
Xu et al. [60]	998	11131	No	No	No	Yes	No	No	20 × 20 × 6, 30 × 30 × 10, 40 × 40 × 26	Yes	Yes	3D	CNN	0.757
Tan et al.. [61]	799	1067	No	No	−1200, 200	0–1	No	Yes	512 × 512 × 32	Yes	No	3D	CNN	0.956
Ozdemir et al. [62]	799	1067	No	No	−1000, 400	0–1	1 × 1 × 1	Yes	64 × 64 × 64	Yes	Yes	3D	V-net	0.921
Masood et al. [63]	799	1067	No	No	No	No	0.5 × 0.5 × 0.5	Yes	Yes	Yes	Yes	3D	Faster R-CNN	0.946
Guo et al. [64]	463	1272	No	Yes	No	No	No	Yes	128 × 128 × 128	No	No	3D	DeepLN	I1: 0.716/I2:0.699
Chenyang et al. [65]	799	1067	No	No	−1000, 400	0–1	1 × 1 × 1	Yes	64 × 64 × 64	Yes	Yes	3D	V-Net	0.902
Cao et al. [66]	799	1067	No	No	No	No	No	Yes	64 × 64 × 3	Yes	Yes	2D/3D	TSCNN	0.911
Xie et al. [67]	710	949	No	No	−1000, 4000	0–1	1 × 1 × 1	Yes	512 × 512	Yes	Yes	2D	Faster R-CNN	0.79
Winkels et al. [68]		38889	No	No	−1000, 300	−1, 1	No	Yes	12 × 72 × 72	Yes	No	3D	3D G-CNNs	0.88
Nasrullah et al. [69]	710	949	No	Yes	No	No	No	Yes	96 × 96 × 96	No	No	3D	CMixNet with faster R-CNN	0.9421
Xia Huang et al. [70]	710	949	No	No	−1000, 3000	0, 255	No	No	No	Yes	Yes	2D	Faster R-CNN	0.88
Wenkai Huang et al. [71]	1699	16312	No	Yes	No	No	No	Yes	96 × 96	Yes	Yes	2D	CNN	0.876
Gong et al. [67]	799	1067	No	Yes	−1200, 600	0, 255	1 × 1 × 1	Yes	128 × 128 × 128	Yes	Yes	3D	RPN	0.904
Zhang et al. [72]	799	1067	No	Yes	No	No	1 × 1 × 1	Yes	32 × 32 × 32	No	Yes	3D	DCNN	0.947
Gu et al. [73]	799	1067	No	Yes	No	No	No	Yes	64 × 64 × 64	No	Yes	3D	Deep CNN	0.7967
Setio et al. [74]	799	1067	No	No	No	No	0.5 × 0.5 × 0.5	No	512 × 512	Yes	No	2D	Znet	0.811

## Data Availability

Data are contained within the article and Appendix A.

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
