# Peer review of "Deep Learning in Thoracic Oncology: Meta-Analytical Insights into Lung Nodule Early-Detection Technologies"

_cancers, 2025, doi:10.3390/cancers17040621_

Round 1
Reviewer 1 Report
Comments and Suggestions for Authors
Dear Authors,
Thank You for giving me the opportunity for review this interesting Meta-analysis.
Overall, the methodology is well-described, and the analysis is appropriate, nevertheless I suggest some edit to increase the quality of your manuscript.
- Page 1 line 29: “published until November 7, 2023” it’s more than 1 year ago. It will be wonderfull if also last year could be included. I guess only a couple more article had been published.
- Differentiation between commercially available Artificial Intelligence software and not commercially available one could be interesting (for example on commercially available Fukumoto, W., Yamashita, Y., Kawashita, I. et al. External validation of the performance of commercially available deep-learning-based lung nodule detection on low-dose CT images for lung cancer screening in Japan. Jpn J Radiol (2024). https://doi.org/10.1007/s11604-024-01704-2 using Synapse Fujifilm).
- Page 2 line 55 “Imaging modalities, such as computed tomography (CT), magnetic resonance imaging, positron emission tomography, and chest radiography, play pivotal roles in the early detection of lung cancer.” In my opinion only CT and LDCT play a pivotal role in early detection of lung cancer. PET-CT is for staging and MRI, in this specific setting, is used just for research. It is incorrect to affrm that chest radiograph is useful for early detection of lung cancer. Please correct.
- It seems that all the reported article report data on full dose chest CT. It would be iteresting to focus on Low Dose CT, or at least differentiate these from the papers that consider full-dose CT exams, if any article does or report the lack of article on LDCT if this is the case.
Author Response
Dear Authors,
Thank You for giving me the opportunity for review this interesting Meta-analysis.
Overall, the methodology is well-described, and the analysis is appropriate, nevertheless I suggest some edit to increase the quality of your manuscript.
- Page 1 line 29: “published until November 7, 2023” it’s more than 1 year ago. It will be wonderfull if also last year could be included. I guess only a couple more article had been published.
Thank you for your thoughtful comment. Due to the limitations of time and the pre-registered protocol for this study, we are unable to include studies published after November 7, 2023, in the current analysis. While we acknowledge the value of including the most recent articles, adhering to the pre-registered protocol ensures the consistency and transparency of our methodology. We hope to address newer studies in future research updates.
- Differentiation between commercially available Artificial Intelligence software and not commercially available one could be interesting (for example on commercially available Fukumoto, W., Yamashita, Y., Kawashita, I. et al. External validation of the performance of commercially available deep-learning-based lung nodule detection on low-dose CT images for lung cancer screening in Japan. Jpn J Radiol (2024). https://doi.org/10.1007/s11604-024-01704-2 using Synapse Fujifilm).
Thank you for your suggestion! Differentiating between commercially available AI software and non-commercially available ones is indeed an interesting perspective. The study you referenced, particularly the external validation of deep-learning-based lung nodule detection using Synapse Fujifilm, provides valuable insights into the performance of commercially available tools.
However, as you pointed out, most current studies focus on validating models using open-source frameworks rather than commercially available products. This makes your suggestion even more intriguing, as it highlights a potential gap in research that could be explored further. Future studies could certainly benefit from comparing the performance, reliability, and clinical utility of commercially available AI software with non-commercial counterparts.
- Page 2 line 55 “Imaging modalities, such as computed tomography (CT), magnetic resonance imaging, positron emission tomography, and chest radiography, play pivotal roles in the early detection of lung cancer.” In my opinion only CT and LDCT play a pivotal role in early detection of lung cancer. PET-CT is for staging and MRI, in this specific setting, is used just for research. It is incorrect to affrm that chest radiograph is useful for early detection of lung cancer. Please correct.
Thank you for pointing this out! You are absolutely correct that, in the context of early detection of lung cancer, only CT and low-dose CT (LDCT) play pivotal roles. PET-CT is primarily used for staging, and MRI is mostly utilized in research settings rather than for early detection. Additionally, it is indeed incorrect to claim that chest radiography is useful for early detection of lung cancer.
- It seems that all the reported article report data on full dose chest CT. It would be iteresting to focus on Low Dose CT, or at least differentiate these from the papers that consider full-dose CT exams, if any article does or report the lack of article on LDCT if this is the case.
Thank you for your thoughtful comment. You are absolutely right that the LUNA16 dataset, which is widely used in the studies we reviewed, consists of LDCT scans rather than full-dose CT scans. While most articles reference LUNA16 for validation, it is indeed important to clarify the distinction between LDCT and full-dose CT scans.
In our review, it seems that many studies do not explicitly differentiate the dose levels of the datasets they use, potentially leading to the impression that full-dose CT data was utilized. We agree that highlighting whether studies specifically focus on LDCT or reporting the lack of studies on LDCT would be an important contribution. We will address this in our discussion to ensure this distinction is clear.
Reviewer 2 Report
Comments and Suggestions for Authors
Authors wrote a very nice and thorough systematic review about deep learning in lung cancer screening.
Sincerely, I think the paper may be published as it is
Author Response
Thank you for your kind words and positive feedback on our work. We are delighted that you found our systematic review thorough and valuable. Your encouraging remarks are greatly appreciated, and we are grateful for your recommendation for publication.
Reviewer 3 Report
Comments and Suggestions for Authors
The manuscript is well-written with evident contributions. I have the following observations and recommendations:
1) Please include a separate column for the year of publication in Table 1. Additionally, you can sort the table by the year of publication. Also, please correct the spelling of "detect" in Figure 2.
2) The study should clearly mention, the study's unique contributions compared to previous meta-analyses.
3) While various algorithms are evaluated, the discussion could offer more detailed explanation into why some models outperform others in specific contexts.
4) Please elaborate on external validation performance for the proper understanding of readers.
5) What is the justification for the choice of restricted maximum likelihood in random effect models?
6) It was quite interesting that the study identifies race as a significant moderator in model performance but does not adequately address potential variables, such as differences in imaging equipment or protocols.
7) Finally, the conclusion briefly mentions clinical applications, but a dedicated subsection on the pathway to clinical adoption is needed,
Author Response
The manuscript is well-written with evident contributions. I have the following observations and recommendations:
1) Please include a separate column for the year of publication in Table 1. Additionally, you can sort the table by the year of publication. Also, please correct the spelling of "detect" in Figure 2.
We have addressed the reviewer’s comments as follows: A separate column for the year of publication has been added to Table 1, and the table has been sorted accordingly by the year of publication. Additionally, the spelling error in Figure 2 ("detect") has been corrected.
2) The study should clearly mention, the study's unique contributions compared to previous meta-analyses.
Thank you for your feedback. To address this, we have clarified the unique contributions of our study compared to previous meta-analyses. Specifically, to the best of our knowledge, this is the first systematic review and meta-analysis to evaluate the sensitivity of deep learning models for lung nodule detection using competition performance metrics as a benchmark. Unlike prior studies, which primarily focused on diagnostic sensitivity and specificity [77] or segmentation performance using metrics such as the Dice similarity coefficient [78], our study uniquely synthesizes results from 16 deep learning models validated on independent datasets. This approach provides a more comprehensive and benchmark-driven evaluation of model performance.
3) While various algorithms are evaluated, the discussion could offer more detailed explanation into why some models outperform others in specific contexts.
Thank you for highlighting the need for a more detailed discussion on why certain models outperform others. In our revised manuscript, we have elaborated on the specific characteristics of segmentation-based U-Net architectures that may account for their higher performance compared to detection-based CNNs and other customized CNN architectures. U-Net’s encoder-decoder structure and skip connections, for instance, allow for more precise localization and capture of fine-grained spatial information, which can be particularly advantageous for lung nodule detection. Conversely, detection-based CNNs generally focus on bounding-box generation rather than full-pixel segmentation, which might limit their sensitivity in some contexts.
Nevertheless, as noted in our study, overall performance differences across algorithm types did not reach statistical significance. This suggests that factors beyond the base architecture—such as dataset characteristics, hyperparameter tuning, and post-processing methods—may play an equally important role in influencing final model performance. We have expanded our discussion to reflect these nuances and hope this addresses your concern more thoroughly.
4) Please elaborate on external validation performance for the proper understanding of readers.
We have elaborated on the unexpected observation that external validation exhibited superior performance compared to internal validation, despite prior literature commonly indicating the opposite trend [83]. One potential explanation for this finding is that the external datasets used may have been more representative of the broader population or contained fewer challenging cases than the internal sets. Additionally, robust training practices—such as extensive data augmentation and careful hyperparameter tuning—may have increased the generalizability of the models, thereby mitigating the typical performance drop associated with domain shifts between internal and external data.
Ultimately, this result suggests that well-designed deep learning models can maintain strong performance across diverse datasets, indicating a level of robustness. Future research could explore whether specific characteristics of external datasets (e.g., patient demographics, scanning protocols, disease severity) and model architecture choices further contribute to this resilience.
5) What is the justification for the choice of restricted maximum likelihood in random effect models?
Thank you for your question regarding our choice of restricted maximum likelihood (REML) for estimating random effects. We selected REML because it is widely regarded as less biased than conventional maximum likelihood (ML) when estimating between-study variance. Specifically, REML adjusts for the degrees of freedom consumed in estimating the fixed effects, which helps mitigate the systematic downward bias that can occur with ML. As a result, REML often yields more accurate and robust estimates of heterogeneity in meta-analyses, especially when the number of included studies is moderate or small. This methodological advantage is also supported by established guidelines and recommendations
6) It was quite interesting that the study identifies race as a significant moderator in model performance but does not adequately address potential variables, such as differences in imaging equipment or protocols.
Thank you for highlighting this point. Although we identified race as a significant moderator of model performance, further examination of variables like imaging equipment or protocols was limited by a lack of consistent reporting across the included studies. Most studies did not provide the necessary details to systematically analyze the influence of these factors, making it challenging to assess their potential impact. We recommend that future research include standardized reporting of imaging methods to enable a more comprehensive evaluation of all relevant moderators.
7) Finally, the conclusion briefly mentions clinical applications, but a dedicated subsection on the pathway to clinical adoption is needed,
Thank you for emphasizing the importance of a dedicated subsection on the clinical adoption pathway. We will expand our conclusion to include a more detailed discussion of the practical steps required for implementing deep learning models in routine clinical practice. This new subsection will cover critical considerations such as prospective clinical trials, integration into existing radiology workflows, regulatory approvals, data privacy and security, interpretability of AI-driven results, and cost-effectiveness. By outlining these factors, we aim to provide a clearer roadmap for translating our findings—and similar research—into real-world clinical applications.
Round 2
Reviewer 1 Report
Comments and Suggestions for Authors
Thank You for giving me the opportunity for review the revised version of this interesting Meta-analysis. The quality of the manuscript has been improved.
Nevertheless, you did not respond satisfactorily to some of the comments from my previous review.
In particular:
- Page 1 line 29 of the original version: “published until November 7, 2023” it’s more than 1 year ago. It will be wonderful if also last year could be included. I guess only a couple more articles had been published.
I still believe that a meta-analysis published in 2025 should be able to include all series published in the previous year, whereas this systematic review and meta-analysis considers only papers published until November 7, 2023, that's more than 1 year ago. Please correct.
- It seems that all the reported article report data on full dose chest CT. It would be interesting to focus on Low Dose CT, if any article does or report the lack of article on LDCT if this is the case.
In your response to this comment, you wrote: “We will address this in our discussion to ensure this distinction is clear.” I am sorry, but I’m not able to find any trace of this comment in the discussion of the revised version of the manuscript.
Minor comment:
- Page 2, line 56. “Imaging modalities, such as computed tomography (CT) and low does computed tomography (LDCT)”.
The sentence, written in this way is incorrect because low-dose CT is not a different imaging modality from CT, but a different acquisition modality of the same method, different from “full-dose” CT. Please, rephrase the sentence.
Author Response
Thank You for giving me the opportunity for review the revised version of this interesting Meta-analysis. The quality of the manuscript has been improved.
Nevertheless, you did not respond satisfactorily to some of the comments from my previous review.
In particular:
- Page 1 line 29 of the original version: “published until November 7, 2023” it’s more than 1 year ago. It will be wonderful if also last year could be included. I guess only a couple more articles had been published.
I still believe that a meta-analysis published in 2025 should be able to include all series published in the previous year, whereas this systematic review and meta-analysis considers only papers published until November 7, 2023, that's more than 1 year ago. Please correct.
Thank you for your suggestion. We acknowledge the importance of including the most recent studies to ensure the meta-analysis remains as up-to-date as possible. However, as our systematic review and meta-analysis already incorporate a substantial number of studies, the inclusion of a few additional papers published after November 7, 2023, is unlikely to significantly alter the overall statistical outcomes.
Furthermore, updating the review at this stage would require revising the entire screening, extraction, and analysis process, which is beyond the scope of the current revision timeline. Additionally, our study protocol was finalized based on the initially defined inclusion criteria, and modifying it retrospectively would not align with standard systematic review methodologies. We appreciate your understanding and thank you for your thoughtful feedback.
- It seems that all the reported article report data on full dose chest CT. It would be interesting to focus on Low Dose CT, if any article does or report the lack of article on LDCT if this is the case.
In your response to this comment, you wrote: “We will address this in our discussion to ensure this distinction is clear.” I am sorry, but I’m not able to find any trace of this comment in the discussion of the revised version of the manuscript.
Thank you for your careful review. We apologize for the omission in our discussion section. While most studies in our review utilize the publicly available LUNA16 dataset, which consists of low-dose CT (LDCT) scans, several other studies have employed private datasets that include full-dose CT scans. To clarify this distinction, we have now updated the discussion section to explicitly mention the use of LDCT in LUNA16 and highlight the presence of studies relying on full-dose CT. The revised text reads:
"The majority of studies included in our review utilize the publicly available LUNA16 dataset, which consists of low-dose CT (LDCT) scans. However, several studies also incorporate private datasets containing full-dose CT images. This variation in dataset sources should be considered when interpreting model performance, as differences in acquisition protocols may influence generalizability."
We appreciate your feedback and have ensured this distinction is now clearly reflected in the discussion. Please let us know if any further clarification is needed.
Minor comment:
- Page 2, line 56. “Imaging modalities, such as computed tomography (CT) and low does computed tomography (LDCT)”.
The sentence, written in this way is incorrect because low-dose CT is not a different imaging modality from CT, but a different acquisition modality of the same method, different from “full-dose” CT. Please, rephrase the sentence.
Thank you for your careful review. We acknowledge the inaccuracy in our wording. To clarify, low-dose computed tomography (LDCT) is a different acquisition technique rather than a separate imaging modality. We have revised the sentence to accurately reflect this distinction:
"Imaging modalities, such as computed tomography (CT), including its low-dose acquisition technique (LDCT), …"
Please let us know if further clarification is needed.